# CasualHDR: Robust High Dynamic Range 3D Gaussian Splatting from Casually Captured Videos

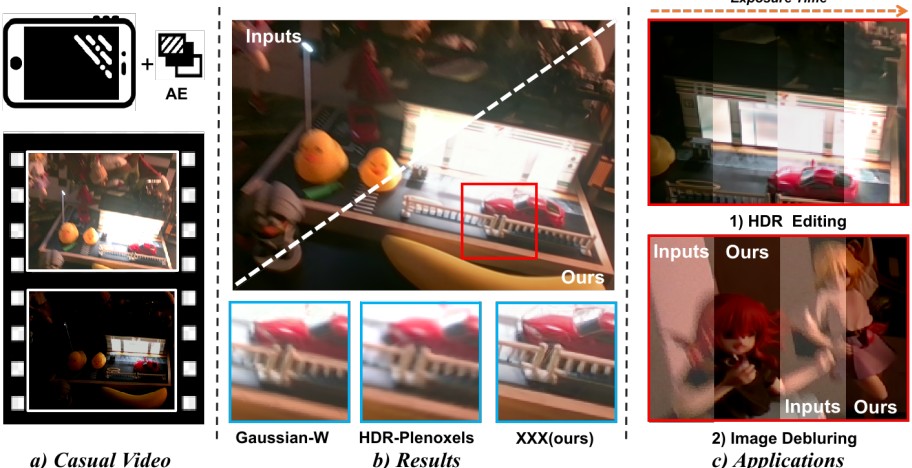

Figure 1: a) Our method can reconstruct 3D HDR scenes from videos casually captured with auto-exposure enabled. b) Our approach achieves superior rendering quality and greater robustness compared to methods like Gaussian-W and HDR-Plenoxels. c) After 3D HDR reconstruction, we can not only synthesize novel view, but also perform various downstream tasks, such as 1) HDR exposure editing, 2) image deblurring.

## Abstract

In recent years, thanks to innovations in 3D scene representation, novel view synthesis and photo-realistic dense 3D reconstruction from multi-view images, such as neural radiance field (NeRF) and 3D Gaussian Splatting (3DGS), have garnered widespread attention due to their superior performance. However, most works rely on low dynamic range (LDR) images and representations of scenes, which limits the capturing of richer scene details. Prior works have focused on high dynamic range (HDR) scene recovery, typically require repeatedly capturing of multiple sharp images with different exposure times at fixed camera positions, which is time-consuming and challenging in practice. For a more flexible data acquisition, we propose a one-stage method: **CasualHDR** to easily and robustly recover the 3D HDR scene from casual videos with auto-exposure (AE) enabled, even in the presence of severe motion blur and varying exposure time. CasualHDR contains a unified differentiable physical imaging model which jointly optimize (i.e. bundle adjust) exposure time, camera response function (CRF), continuous-time camera motion trajectory on $\mathbb{SE}(3)$, and the 3DGS-based HDR scene. Extensive experiments demonstrate that our approach outperforms existing reconstruction methods in terms of robustness and rendering quality. Three applications can be achieved after the 3DGS HDR scene reconstruction: novel-view synthesis, image deblurring (deblur input images) and HDR editing (adjust the exposure time thus brightness of the input images).

## 1 Introduction

Photo-realistic 3D scene reconstruction and Novel View Synthesis (NVS) are essential areas in computer vision with applications in VR/AR, autonomous driving, and embodied AI, offering immer-

sive experiences for both humans and AI agents. Neural Radiance Fields (NeRFs) (Mildenhall et al., 2020) have become a mainstream approach in NVS due to their high-quality rendering. The introduction of 3D Gaussian Splatting (3DGS) (Kerbl et al., 2023) further advanced the field. In contrast to implicit representations like NeRFs, 3DGS uses explicit 3D Gaussian primitives, greatly improving training and rendering efficiency yielding high-quality images, making it a popular choice.

However, most 3D reconstruction methods struggle with high-contrast inputs, assuming good-quality images with consistent exposure conditions and low dynamic range (LDR). Limited dynamic range of the inputs hinders 3D scene representations from reconstructing fine details in high dynamic range (HDR) environments, thereby restricting its further applications, e.g., 3D HDR content creation. Although 2D HDR contents (i.e. images and videos) have been standardized, consumed and exploited in recent years (Hannuksela et al., 2015; ITU-R, 2018; Alakuijala et al., 2019), 3D HDR free-viewpoint (volumetric) content is still a new concept with great potential value. Therefore, reconstructing high dynamic range (HDR) scenes is of significant practical value for achieving better visual effects and meeting the needs of downstream tasks.

Current 3D HDR reconstruction methods can be divided into two categories. The first category, e.g. RawNeRF (Mildenhall et al., 2022) and LE3D (Jin et al., 2024) etc., takes in noisy RAW images, aiming to reconstruct noise-free 3D HDR scenes. The second category, represented by HDR-NeRF (Huang et al., 2022) , HDR-GS (Cai et al., 2024), HDR-Plenoxels (Jun-Seong et al., 2022) and Cinematic Gaussians (Wang et al., 2024a), draws inspiration from HDR imaging (HDRI), using multi-exposure LDR images at fixed positions as inputs to reconstruct the 3D HDR scene while learning camera response function (CRF). However, the strict inputs and high reconstruction costs limit their flexibility and broader applications. The challenges include: 1) Data acquisition of RAW images and accurate exposure time is ususally expensive due to the use of professional equipment.; 2) In low-light conditions, long exposure times increase the risk of motion blur from camera shake, reducing reconstruction quality; 3) The geometric consistency will be compromised if given inaccurate camera pose initialization, as the camera poses are not being optimized. Thus, a key challenge is reducing the cost of data acquisition, enabling high-quality 3D HDR scene reconstruction with consumer-grade devices.

Most modern consumer-grade cameras use auto-exposure during video recording, automatically adjusting exposure time based on ambient lighting. This expands the captured dynamic range in the video, making it possible for us to reconstruct 3D HDR scenes. However, naively applying these videos to existing HDR 3D reconstruction methods presents several challenges: 1) Accurate exposure times for individual frames are often unknown; 2) Auto-exposure can cause inconsistencies in brightness between frames, leading to pose estimation errors in structure from motion (SfM) frameworks; 3) In low-light conditions, longer exposure times combined with camera movement during recording often cause severe motion blur.

To address these challenges, we cannot assume that the camera is static during the exposure time as in previous methods. Therefore, we must also account for camera motion during this period. Through analyzing the physical imaging process, we found that both motion blur and brightness variations are both directly related to the exposure time. For example, a longer exposure time lead to more severe motion blur and higher image brightness. Thus, camera motion blur can serve as an indicator of the exposure time, providing a useful constraint for joint optimization.

Building on above reasoning, we propose a one-stage method called **CasualHDR**, which is an unified 3DGS-based HDR reconstruction framework that couples the physical imaging model with camera motion representation, impoving the robustness and flexibility. In our designed unified imaging model, the continuous-time camera trajectory on $\mathbb{SE}(3)$, exposure time, and camera response function are jointly optimized and mutually constrained. Therefore, our approach does not require ground truth exposure times as previous methods.

Our method takes as input **a casual video** captured by a consumer-grade imaging device, where each frame exhibits brightness variations and motion blur due to different unknown exposure times. To evaluate the effectiveness of our method, we conducted experiments using synthetic datasets generated by Blender and self-captured real-world datasets. The results demonstrate that our method outperforms other approaches in 3D HDR reconstruction, achieving high-quality rendering.

In summary, our **contributions** can be outlined as follows:

- **CasualHDR**, a unified imaging model that jointly optimizes continuous-time camera trajectory, CRF, exposure times and 3DGS-based HDR representation, which enables users to reconstruct 3D HDR scenes from casually captured videos at a low cost.

- A dataset that includes both synthetic data and real-world data, where each video contains severe variations in brightness and camera motion blur, that can be useful to the community to further investigate into this problem.

- With extensive experiments, we demonstrate how to utilize this model to reconstruct high-quality HDR scenes from casual videos, and exhibit state-of-the-art performance across all datasets.

## 2 RELATED WORK

### 2.1 HIGH DYNAMIC RANGE IMAGING

High Dynamic Range Imaging (HDRI) enhances luminosity beyond standard digital imaging by merging multi-exposure LDR images from fixed poses. In video capture, alternating long and short exposures achieves similar effects. Recently, deep learning approaches have treated HDRI as an image domian translation task, designing networks to convert LDR to HDR images. However, camera disturbances often lead to ghosting artifacts. To address this, Gryaditskaya et al. (2015) proposed an adaptive metering algorithm to adjust exposure and reduce motion artifacts, while other methods use spatial attention to mitigate motion blur. With the advent of 3D scene representations such as NeRF and 3DGS, methods like (Huang et al., 2022; Jun-Seong et al., 2022; Cai et al., 2024; Huang et al., 2024; Wang et al., 2024a) have emerged to reconstruct 3D HDR scenes and calibrate CRFs simultaneously. While these methods are effective, they often rely on precise exposure times and struggle with motion blur, highlighting the need for improved robustness and generalizability.

### 2.2 IMAGE DEBLURRING

Image deblurring aims to restore sharp images from blurred ones, and current techniques are categorized into three main types. The first type uses hand-crafted priors, such as total variation and heavy-tailed gradient priors, to constrain the solution space, solving for the blur kernel. However, these methods are limited as different blur kernels can produce similar blurred effects (Krishnan & Fergus, 2009; Cho & Lee, 2009). The second type is deep learning-based, which achieves end-to-end image restoration by training on large datasets, with notable methods including MPRNet (Zamir et al., 2021) and Stripformer (Tsai et al., 2022). Despite their success, these 2D approaches sometimes struggle with tasks requiring multi-view geometric consistency. The third type utilizes multi-view blurred images to reconstruct the 3D scene representation and deblur the images while adhering to geometric constraints. Pioneering works in this category include Deblur-NeRF (Ma et al., 2021), PDRF (Peng & Chellappa, 2022), DP-NeRF (Lee et al., 2023a) and BAD-NeRF (Wang et al., 2023). Deblur-NeRF, PDRF and DP-NeRF jointly learn blur kernels with the radiance field to approximate the blurring process, while BAD-NeRF proposed a physical motion blur imaging model that jointly recovers (i.e. bundle adjusts) the radiance field along with camera trajectories on $\mathbb{SE}(3)$. Following BAD-NeRF, numerous emerging works (Lee et al., 2023b; Li et al., 2024a; Lee et al., 2024b;a; Sun et al., 2024; Chen & Liu, 2024; Oh et al., 2024; Zhao et al., 2024; Yu et al., 2024; Li et al., 2024b; Qi et al., 2024; Tang et al., 2024) have proved the effectiveness of continuous $\mathbb{SE}(3)$ trajectory representations for modeling coupled camera motion and imaging characteristics in the process of joint 3D reconstruction and multi-view image recovery.

### 2.3 ROBUST NOVEL VIEW SYNTHESIS

Novel view synthesis involves generating images from arbitrary viewpoints using a series of images with known poses. Neural Radiance Fields (NeRF) has significantly advanced the field by reconstructing the radiance field as the 3D scene representation to render images from new perspectives. Building on NeRF, 3D Gaussian Splatting (3DGS) was proposed, which uses explicit Gaussian primitives to represent the 3D scene, significantly improving training and rendering speeds while maintaining good image quality. Most novel view synthesis methods assume high-quality input data; however, when this assumption is violated—such as with blurry images, large exposure variations,

or inaccurate poses—the reconstruction quality degrades rapidly, producing artifacts.To address this, NeRF-W (Martin-Brualla et al., 2021) and Gaussian-W (Zhang et al., 2024) attach an optimizable appearance vector to each image, modeling varying appearances from internet-sourced images. HDR-NeRF (Huang et al., 2022) reconstructs HDR 3D scenes using multi-view, multi-exposure images with known precise exposure times, while HDR-HexPlane (Wu et al., 2024) extends this to dynamic scenes, enabling fast reconstruction even with unknown exposure times. Other approaches, such as Fu et al. (2024), tackle reconstruction with inaccurate or random poses, reducing reliance on traditional SfM methods. Methods like Zhao et al. (2024) incorporate camera motion models to handle blurry inputs, achieving deblurring while reconstructing the 3D scene. $I^2$-SLAM(Bae et al., 2024) is a concurrent work similar to ours, capable of using images with exposure inconsistencies and blur as input. However, it focuses on RGB-D SLAM, and its representations of trajectory and CRF differ from ours; Meanwhile, it cannot adjust the rendering exposure times fexibly as ours, as its CRF module follows Jun-Seong et al. (2022). These approaches improve robustness in handling various forms of image degradation, enhancing the quality of novel view synthesis.

## 3  METHOD

In this section, we will provide a detailed explanation of our proposed **CasualHDR**, which takes video captured with auto-exposure settings as input. In Section 3.1, we will first give a brief overview of the scene representation based on 3D Gaussian Splatting, followed by a description of the camera's continuous motion trajectory in Section 3.2. Section 3.3 will detail the camera imaging model and explain how we integrate exposure time to link both components. Finally, we will introduce the loss functions used in Section 3.4. A detailed illustration of the method is provided in Figure 2. We will now elaborate on each component.

### 3.1  PRELIMINARY: 3D GAUSSIAN SPLATTING

3D-GS represents the scene as 3D Gaussian primitives denoted as $\mathbf{G}$. Each 3D Gaussian primitive is characterized by a mean position $\boldsymbol{\mu} \in \mathbb{R}^3$, opacity $\mathbf{o} \in \mathbb{R}$, color $\mathbf{c} \in \mathbb{R}^3$, and a 3D covariance matrix $\boldsymbol{\Sigma} \in \mathbb{R}^{3 \times 3}$. To ensure $\boldsymbol{\Sigma}$ remains positive semi-definite, it is parameterized using a scaling matrix $\mathbf{S} \in \mathbb{R}^3$ and a rotation matrix $\mathbf{R} \in \mathbb{R}^{3 \times 3}$, which is stored as a quaternion $\mathbf{q} \in \mathbb{R}^4$. During rendering, the 3D Gaussians are projected onto the image plane at a specific pose $\mathbf{P}_i$, transforming $\boldsymbol{\Sigma}$ into a 2D covariance matrix $\boldsymbol{\Sigma}' \in \mathbb{R}^{2 \times 2}$. These can be mathematically expressed as:

$$\mathbf{G}(\mathbf{x}) = e^{-\frac{1}{2}(\mathbf{x}-\boldsymbol{\mu})^\top \boldsymbol{\Sigma}^{-1}(\mathbf{x}-\boldsymbol{\mu})}, \qquad \boldsymbol{\Sigma} = \mathbf{R}\mathbf{S}\mathbf{S}^T\mathbf{R}^T, \qquad \boldsymbol{\Sigma}' = \mathbf{J}\mathbf{R}_c\boldsymbol{\Sigma}\mathbf{R}_c^T\mathbf{J}^T, \qquad (1)$$

where $\mathbf{J} \in \mathbb{R}^{2 \times 3}$ is the Jacobian of the affine approximation of the projective transformation. Next, the 2D Gaussians undergo depth sorting followed by tile-based rasterization. The final color values for individual pixels are obtained using $\alpha$-blending:

$$\mathbf{C}(x, y, \mathbf{P}_i) = \sum_{i=1}^{N} \mathbf{c}_i \alpha_i \prod_{j=1}^{i-1}(1 - \alpha_j), \quad \alpha_i = \mathbf{o}_i \cdot \exp(-\sigma_i), \quad \sigma_i = \frac{1}{2}\Delta_i^T \boldsymbol{\Sigma}'^{-1}\Delta_i, \qquad (2)$$

where $\mathbf{c}_i$ is the learnable color of each Gaussian, and $\alpha_i$ is the alpha value determined by the 2D covariance $\boldsymbol{\Sigma}'$ multiplied by the learned Gaussian opacity $\mathbf{o}$. $\Delta_i \in \mathbb{R}^2$ represents the offset between the pixel center and the 2D Gaussian center. The above derivations show that the rendered pixel color, $\mathbf{C}$ in Eq. (2), is differentiable with respect to all learnable Gaussian parameters $\mathbf{G}$ and camera poses $\mathbf{P}$, which is crucial for our bundle adjustment formulation and allows incorporating motion-blurred images and inaccurate camera poses into the 3D-GS framework.

### 3.2  CONTINUOUS TRAJECTORY REPRESENTATION

Cumulative $\mathbb{SE}(3)$ B-spline is a widely used continuous-time trajectory representation in robotics, especially in state estimation, sensor fusion and path planning (Furgale et al., 2012; Lovegrove et al., 2013; Bry et al., 2015; Rehder et al., 2016; Mueggler et al., 2018; Geneva et al., 2020) because of many excellent characteristics: such as $C^2$ continuity, locality and convex hull property that delicately incorporates gradient information and dynamic constraints, which converges quickly to generate smooth and feasible trajectories (Zhou et al., 2019) . $\mathbb{SE}(3)$ B-spline allows for the calculation of pose, velocity, and accelerations at any timestamp given along a trajectory.

Figure 2: **The pipeline of CasualHDR.** Given a casually captured video with auto exposure, camera motion blur, and significant exposure time changes, we train 3DGS to reconstruct an HDR scene. We design a unified model based on the physical image formation process, integrating camera motion blur and exposure-induced brightness variations. This allows for the joint estimation of camera motion, exposure time, and camera response curve while reconstructing the HDR scene. After training, our method can sharpen the train images and render HDR and LDR images from specified poses.

Targeting at unordered inputs, existing multi-view deblurring methods following BAD-NeRF (Wang et al., 2023) model the camera motion and estimate short splines for each frame separately, thus cannot utilize the cross-frame motion constrains and priors, given a continuous video as their input. Some methods (Wang et al., 2021; Li et al., 2022; Sun et al., 2024; Li et al., 2024c; Lin et al., 2024; Shih et al., 2024; Wang et al., 2024b) utilize basis functions to regularize continuous-time deformations to reconstruct dynamic scenes, but robust reconstruction from casual videos with a continuous-time camera trajectory representation has not yet been explored. To this end, this paper estimates the camera motion across the whole video with a continuous-time cumulative $\mathbb{SE}(3)$ B-spline trajectory.

Following Lovegrove et al. (2013), given a series of temporally uniformly distributed control knots, the pose $\mathbf{P}(t)$ at a given timestamp $t$ can be interpolated with 4 adjacent control knots, denoted as $\mathbf{T}_0, \mathbf{T}_1, \mathbf{T}_2$ and $\mathbf{T}_3 \in \mathbb{SE}(3)$:

$$\mathbf{P}(t) = \mathbf{T}_0 \cdot \prod_{j=0}^{2} \exp(\tilde{\mathbf{B}}(u)_{j+1} \cdot \mathbf{\Omega}_j), \quad \tilde{\mathbf{B}}(u) = \mathbf{C} \begin{bmatrix} 1 \\ u \\ u^2 \\ u^3 \end{bmatrix}, \quad \mathbf{C} = \frac{1}{6} \begin{bmatrix} 6 & 0 & 0 & 0 \\ 5 & 3 & -3 & 1 \\ 1 & 3 & 3 & -2 \\ 0 & 0 & 0 & 1 \end{bmatrix} . \quad (3)$$

where $\tau$ represents the spline sampling interval, $u = \frac{t}{\tau}$, and $u$ lies within the interval $[0, 1)$; $\tilde{\mathbf{B}}(u)_{j+1}$ denotes the $(j+1)^{th}$ element of the vector $\tilde{\mathbf{B}}(u)$, $\mathbf{\Omega}_j = \log(\mathbf{T}_j^{-1} \cdot \mathbf{T}_{j+1})$, based on the Qin (1998).

## 3.3 PHYSICAL IMAGE FORMATION MODEL

The physical image formation process refers to a digital camera collecting scene irradiance during the exposure time $\Delta t$ and converting them into measurable electric charges, which are ultimately mapped into pixel values through the *camera response function* (CRF) defined by $F$. Assuming the camera moves along a continuous trajectory $t \mapsto \mathbf{P}(t)$ during exposure time $\Delta t$ with constant velocity, this process can be mathematically modeled as follows:

$$\mathbf{B}(x, y) = F \left( \int_{t_b}^{t_b + \Delta t} \mathbf{H}(x, y, \mathbf{P}(t)) \, dt \right) \quad (4)$$

where $\mathbf{B}(x, y) \in \mathbb{R}^{\mathrm{H} \times \mathrm{W} \times 3}$ denotes the real captured image, $x, y \in \mathbb{R}^2$ represents the pixel location, $t_b$ denotes the timestamp when the shutter opens, $\mathbf{H}(x, y, \mathbf{P}(t))$ represents scene irradiance mapped into camera at pose $\mathbf{P}(t)$ which is interpolated from the continuous trajectory. Additionally, if the camera moves during the exposure time, the camera will collect irradiance from different scene points, resulting in camera motion blur. The integral part in Eq. (4) can be discretized as follows:

$$\mathbf{H}(x, y) \approx \sum_{k=0}^{N-1} \mathbf{H}_k\left(x, y, \mathbf{P}(t_k)\right) \Delta t_k \approx \frac{1}{N} \sum_{k=0}^{N-1} \mathbf{H}_k\left(x, y, \mathbf{P}(t_k)\right) \Delta t \tag{5}$$

$\mathbf{H}(x, y) \in \mathbb{R}^{\mathrm{H} \times \mathrm{W} \times 3}$ denotes blur HDR image, N represents the number of virtual latent sharp images, $\Delta t_k$ represents the exposure time of virtual camera $k$ and can be set as a constant equal to $\frac{\Delta t}{N}$, $t_k$ denotes the timestamp corresponding to virtual camera $k$, it can be calculated as $t_b + \frac{\Delta t}{N} * k$.

After obtaining $\mathbf{H}(x, y)$, we need to use the camera response function $F$, which includes image-varying white balance WB and tone mapping TM, to convert it into an LDR image:

$$\mathbf{B}(x, y) = \mathbf{F}(\mathbf{H}(x, y)) = \mathrm{TM} \circ \mathrm{WB}(\mathbf{H}(x, y)), \mathrm{WB}(\mathbf{c}) = [wb_r, wb_g, wb_b]^T \odot [c_r, c_g, c_b]^T . \tag{6}$$

Due to the fact that RGB channels have different camera response curves for TM, we adopt separate MLP for each channel. Unlike prior methods, we treat $\Delta t$ as an optimizable quantity rather than a precisely known parameter. Initially, $\Delta t$ can be assigned **a random value**. Since the exposure time directly affects the brightness and motion blur of the image, it will be gradually optimized to the actual value during the subsequent deblurring and HDRI processes. This significantly reduces the dependency on the exposure time and enhances the robustness of 3D HDR reconstruction.

## 3.4 LOSS FUNCTION

Given a series of video frames moving along a continuous trajectory, we can estimate the learnable Gaussian primitives, the camera trajectory parameters, implicit CRF representation and the exposure time for each image. This estimation can be achieved by minimizing a loss function, which can be specifically expressed as follows:

$$\mathcal{L} = \mathcal{L}_{\mathrm{rec}} + \lambda_{\mathrm{exp}} \mathcal{L}_{\mathrm{exp}}, \qquad \mathcal{L}_{\mathrm{rec}} = (1 - \lambda)\mathcal{L}_1 + \lambda \mathcal{L}_{\mathrm{D\text{-}SSIM}}, \tag{7}$$

where $\mathcal{L}_{\mathrm{rec}}$ can constrain the consistency between the rendered image $\mathbf{C}_k(\mathbf{x})$ (the $k^{th}$ blurry LDR image synthesized from 3D-GS using the aforementioned image formation model (Eq. 5)) and the input LDR image $\mathbf{C}_k^{gt}(\mathbf{x})$.

To accurately model significant exposure variations in the input images, the second term of the loss function normalizes the images to a medium exposure by scaling pixel intensities before computing discrepancies Liu et al. (2020); Wang et al. (2024a):

$$\mathcal{L}_{\mathrm{exp}} = \mathcal{L}_1\left(\frac{\mathbf{C}_k^{gt}(\mathbf{x})}{\bar{\mathbf{C}}_k^{gt}(\mathbf{x})}, \frac{\mathbf{C}_k(\mathbf{x})}{\bar{\mathbf{C}}k(\mathbf{x})}\right) + \mathcal{L}_{\mathrm{D\text{-}SSIM}}\left(\frac{\mathbf{C}_k^{gt}(\mathbf{x})}{\bar{\mathbf{C}}_k^{gt}(\mathbf{x})}, \frac{\mathbf{C}_k(\mathbf{x})}{\bar{\mathbf{C}}_k(\mathbf{x})}\right), \tag{8}$$

where $\bar{\mathbf{C}}_k^{gt}(\mathbf{x})$ and $\bar{\mathbf{C}}_k(\mathbf{x})$ represent the average pixel value of $\mathbf{C}_k^{gt}(\mathbf{x})$ and $\mathbf{C}_k(\mathbf{x})$. We set $\lambda_{\mathrm{exp}} = 0.25$ in all our experiments, and train our models using the Adam optimizer (Kingma & Ba, 2017).

## 4 EXPERIMENTS

### 4.1 DATASETS

**Synthetic datasets.** We generated a synthetic dataset using Blender 3.6 with the Cycles engine, featuring four distinct scenes: *Factory*, *Pool*, *Cozyroom*, and *Trolley*. Each scene contains 77 images, with manually crafted Bézier camera trajectories. The dataset generation combines physical motion blur imaging model (Wang et al., 2023) and tone mapping from HDR to LDR. For each scene, images were assigned random exposure times, and captured using a continuous camera trajectory, generating sharp HDR images, which were averaged over the exposure period to create motion-blurred images. These HDR blurred images were then processed using the tone-mapping function from HDR-NeRF (Huang et al., 2022) to generate the corresponding LDR blurred images.

**Real datasets.** Since current HDRI datasets consist of multiple images with known exposure times captured from fixed viewpoints, which differs from our approach of using casual videos for HDR scene reconstruction, we captured a challenging real-world dataset, *CasualVideo*, using the Intel RealSense D455 and Google Pixel 8 Pro mounted on a DJI RS3 Mini gimbal. The dataset comprises two subsets: *RealSense* and *Smartphone*. *RealSense* contains four sequences: Yakitori, Toufu, Toufu-vicon and Girls-vicon, where the the latter two sequences have ground truth camera poses from the Vicon motion capture system. *Smartphone* contains two sequences: Building and Fish.

Due to the RealSense camera cannot provide the current exposure times when auto-exposure is enabled, which is detrimental for baseline methods that require exposure time, we implemented our own auto-exposure control with fixed aperture and gain (ISO) following Su & Kuo (2015) on both camera devices. We also developed scripts to extract measured exposure times from the hardware as ground truth labels. Additionally, we utilized the publicly available dataset, ScanNet (Dai et al., 2017), which contains scenes recorded with the auto-exposure feature enabled (Bae et al., 2024), to evaluate the performance of our method on real-world data.

## 4.2 IMPLEMENTATION DETAILS

We implemented our method using PyTorch within the `gsplat` framework (Ye et al., 2024) with MCMC strategy (Kheradmand et al., 2024). The optimization of HDR scene representation, implicit CRF representation, camera motion trajectory, and exposure times was performed using the Adam optimizer, with the learning rate for the Gaussian primitives kept consistent with gsplat. To balance performance and efficiency, we set the number of virtual camera poses (i.e., n in Eq. 5) to 10. For initialization, we used HLoc (Sarlin et al., 2019) instead of COLMAP Schönberger & Frahm (2016) like in many other works to initialize camera poses and Gaussian primitives for the synthetic and our *Realsense* datasets, because we discover that learning-based SfMs performs more robustly in our challenging setting, as the change of exposure time breaks the photo-consistency across consecutive frames, meanwhile, overexposure and underexposure are challenging for hand-crafted feature detection. For the ScanNet dataset (Dai et al., 2017) and our *Smartphone* datasets, DPV-SLAM (Lipson et al., 2024) was used since HLoc (Sarlin et al., 2019) was unable to initialize due to the poor image quality of the scenes. Synthetic dataset experiments were conducted on a single NVIDIA RTX 3090 (24G VRAM) GPU, while real dataset experiments were performed on a single NVIDIA RTX A6000 (48G VRAM) GPU because the real datasets contain more images and require larger VRAM.

## 4.3 BASELINE METHODS AND EVALUATION METRICS

To evaluate the robustness of our method in learning accurate scenes representation under poorly exposed conditions and server motion blur, we compared it against scene reconstruction methods that handle brightness variations, e.g. HDR-NeRF (Huang et al., 2022), HDR-Plenoxels (Jun-Seong et al., 2022), Gaussian-W (Zhang et al., 2024), as well as method for scene reconstruction from blurred images, such as BAD-Gaussians (Zhao et al., 2024). In addition, 3D-GS (Kerbl et al., 2023) implemented by gsplat (Ye et al., 2024) was included as the comparison baseline. The quality of images rendered from the learned scene is evaluated with commonly used metrics such as PSNR, SSIM (Wang et al., 2004), and LPIPS (Zhang et al., 2018). Furthermore, to evaluate whether our method effectively recovers camera motion trajectories, we compared it with pose estimation method, e.g. HLoc (Sarlin et al., 2019), DPV-SLAM (Lipson et al., 2024), BAD-Gaussians (Zhao et al., 2024).For pose estimation accuracy, we utilize absolute trajectory error (ATE) with *mean* and *std* as the metric. $I^2$-SLAM (Bae et al., 2024) is a concurrent work similar to our settings, but since it is not open-sourced, we cannot compare our method against it.

## 4.4 QUANTITATIVE EVALUATION RESULTS.

We conducted experiments with our method using two different settings: one with randomly initialized exposure times (**CasualHDR-random**) and one with ground truth exposure times (**CasualHDR-gt**). We demonstrated the performance of our method in scene learning compared to prior methods through novel view synthesis and image deblurring tasks, while also comparing the ATE metric in the pose estimation task. The results of Scannet dataset and 2 scenes of *Realsense* dataset will be presented in the supplementary materials.

Due to the fact that most images in real-world datasets are blurry, we select 5 to 10 sharp images for each sequence to evaluate metric.The experimental results in Table 1 and Table 2 demonstrate

Figure 3: **Qualitative results of HDR editing with various exposure times.** After reconstruction, **Casual-HDR** can generate any expected exposure time at a given camera pose.

Table 1: **Quantitative comparisons on the synthetic datasets in terms of novel view**

| | Factory | | | Pool | | | Trolley | | | Cozyroom | | |
|---|---|---|---|---|---|---|---|---|---|---|---|---|
| | PSNR↑ | SSIM↑ | LPIPS↓ | PSNR↑ | SSIM↑ | LPIPS↓ | PSNR↑ | SSIM↑ | LPIPS↓ | PSNR↑ | SSIM↑ | LPIPS↓ |
| gsplat (Kerbl et al., 2023) | 15.14 | 0.75 | 0.25 | 11.73 | 0.65 | 0.30 | 14.48 | 0.62 | 0.32 | 13.86 | 0.76 | 0.22 |
| Gaussian-W (Zhang et al., 2024) | 23.68 | 0.75 | 0.26 | 23.28 | 0.69 | 0.62 | 17.83 | 0.64 | 0.34 | 27.16 | 0.85 | 0.15 |
| BAD-Gaussians (Zhao et al., 2024) | 14.99 | 0.81 | 0.25 | 25.09 | 0.72 | 0.30 | 16.12 | 0.66 | 0.22 | 17.12 | 0.75 | 0.20 |
| HDR-Plenoxels (Jun-Seong et al., 2022) | 24.36 | 0.72 | 0.29 | 30.84 | 0.81 | 0.33 | 17.05 | 0.55 | 0.42 | 28.13 | 0.81 | 0.13 |
| HDR-NeRF (Huang et al., 2022) | 14.57 | 0.31 | 0.68 | - | - | - | - | - | - | 13.62 | 0.32 | 0.77 |
| CasualHDR-random (ours) | 30.25 | 0.89 | 0.10 | 32.63 | 0.91 | 0.09 | 25.14 | 0.81 | 0.24 | 29.62 | 0.86 | 0.10 |
| CasualHDR-gt (ours) | 30.75 | 0.90 | 0.09 | 32.36 | 0.92 | 0.08 | 25.85 | 0.88 | 0.11 | 31.32 | 0.92 | 0.09 |

that our method significantly outperforms prior methods in novel view synthesis. Despite using randomly initialized exposure times, **CasualHDR-random** still exceeds previous works due to its ability to jointly optimize exposure times and CRF representation. Unlike HDR-NeRF (Huang et al., 2022), our method can learn accurate HDR scene representations from degraded images without measured exposure times. Note that HDR-NeRF failed in all scenes on the real dataset. Additionally, by modeling the physical principles of actual camera imaging and integrating this into the scene learning, our method shows improved performance over HDR-Plenoxels (Jun-Seong et al., 2022) and Gaussian-W (Zhang et al., 2024). Furthermore, our method utilizes spline representations to optimize camera motion trajectories, facilitating proper scene representation learning, whereas the aforementioned methods struggle without ground truth camera poses.

Table 3 shows that our method achieves superior performance in image deblurring task compared to BAD-Gaussians (Zhao et al., 2024). This is because our method can recover accurate scene representation from images affected by both motion blur and poor exposure.

The experimental results presented in Table 4 demonstrate that our method outperforms prior approaches in the pose estimation task. HLoc, which relies on feature point matching, exhibits poor performance under conditions of varying brightness and motion blur. Although DPV-SLAM (Lipson et al., 2024) and BAD-Gaussians (Zhao et al., 2024) can operate effectively in the presence of motion blur, they struggle to tolerate environments with high-contrast and varying exposure time. This indicates that our method can robustly estimate continuous camera trajectories under high-contrast environments, within varying exposure time and motion blur.

## 4.5 QUALITATIVE EVALUATION RESULTS.

The results in Figure 3 demonstrate that our method can accurately learn HDR scenes and the brightness of the rendered images can be adjusted by manually changing the exposure time. The qualitative comparisons of the NVS and deblurring tasks on both synthetic and real datasets are shown in Figure 4, Figure 5, Figure 6 and Figure 7. The experimental results indicate that our method outperforms previous approaches and is visually closer to the ground truth. This demonstrates that our method can effectively learn scene representations from images that simultaneously exhibit varying

Table 2: **Quantitative comparisons on the real-world datasets in terms of novel view.**

| | Fish-pixel8pro | | | Building-pixel8pro | | | Toufu-vicon | | | Girls-vicon | | |
|---|---|---|---|---|---|---|---|---|---|---|---|---|
| | PSNR↑ | SSIM↑ | LPIPS↓ | PSNR↑ | SSIM↑ | LPIPS↓ | PSNR↑ | SSIM↑ | LPIPS↓ | PSNR↑ | SSIM↑ | LPIPS↓ |
| gsplat | 23.20 | 0.82 | 0.16 | 25.99 | 0.81 | 0.11 | 24.34 | 0.81 | 0.28 | 23.81 | 0.77 | 0.28 |
| BAD-Gaussians | 24.28 | 0.78 | 0.14 | 26.93 | 0.82 | 0.11 | 24.22 | 0.82 | 0.24 | 23.95 | 0.77 | 0.28 |
| HDR-Plenoxels | 19.39 | 0.53 | 0.65 | 26.87 | 0.81 | 0.15 | 17.90 | 0.51 | 0.69 | 26.73 | 0.84 | 0.30 |
| Gaussian-W | 26.13 | 0.83 | 0.15 | 27.99 | 0.82 | 0.11 | 26.38 | 0.83 | 0.29 | 26.88 | 0.86 | 0.25 |
| CasualHDR-random (ours) | 28.30 | 0.83 | 0.13 | 28.79 | 0.83 | 0.09 | 30.87 | 0.90 | 0.15 | 32.00 | 0.90 | 0.19 |
| CasualHDR-gt (ours) | 30.81 | 0.87 | 0.12 | 29.71 | 0.85 | 0.08 | 31.34 | 0.92 | 0.12 | 32.39 | 0.91 | 0.17 |

Table 3: **Quantitative comparisons on the synthetic datasets in terms of deblur**

| | Factory | | | Pool | | | Trolley | | | Cozyroom | | |
|---|---|---|---|---|---|---|---|---|---|---|---|---|
| | PSNR↑ | SSIM↑ | LPIPS↓ | PSNR↑ | SSIM↑ | LPIPS↓ | PSNR↑ | SSIM↑ | LPIPS↓ | PSNR↑ | SSIM↑ | LPIPS↓ |
| BAD-Gaussians (Zhao et al., 2024) | 24.32 | 0.73 | 0.12 | 25.87 | 0.79 | 0.23 | 19.06 | 0.62 | 0.19 | 23.37 | 0.79 | 0.11 |
| CasualHDR-random (ours) | 31.20 | 0.88 | 0.05 | 32.95 | 0.87 | 0.10 | 23.65 | 0.69 | 0.12 | 29.60 | 0.84 | 0.05 |
| CasualHDR-gt (ours) | 32.00 | 0.91 | 0.07 | 34.53 | 0.96 | 0.05 | 29.35 | 0.87 | 0.08 | 33.01 | 0.93 | 0.04 |

Table 4: **Quantitative comparisons for pose estimation on the *Realsense* sequences with Vicon motion captured groundtruth.** The results are in the absolute trajectory error metric (ATE) with units in centimeters.

| | HLoc | DPV-SLAM | BAD-Gaussians | CasualHDR-random (ours) | CasualHDR-gt (ours) |
|---|---|---|---|---|---|
| Toufu-vicon | .4644±.3921 | .4043±.3877 | .3935±.4212 | .3687±.3874 | .3595±.3462 |
| Girls-vicon | 1.528±1.011 | .9557±.8231 | .8548±.8628 | .8294±.8834 | .6478±.8268 |

exposure time and motion blur, while prior work lacks robustness given the challenging conditions and failed to reconstruct high-quality HDR 3D scene.

### 4.6 ABLATION STUDIES.

We conduct experiments to evaluate the performance of our method under various configurations on three different sequences of synthetic datasets(e.g. Pool, Factory and Cozyroom).

**Initialization for camera motion spline.** In our method, the camera motion spline needs to be initialized by leveraging the poses estimated from HLoc (Sarlin et al., 2019) or DPV-SLAM (Lipson et al., 2024) before being optimized. Therefore, the configuration of initialization will impact the performance of our method. We define a *ratio* representing the number of control knots of spline divided by the number of input images , and evaluate the effect of the *ratio*. The results in Table 5 indicate that model performance improves until it saturates as the *ratio* increases. We set *ratio* = 3.0 for all experiments to ensure a trade-off between the performance and computational overhead.

Table 5: **Ablation studies on the $ratio$ for initializing camera motion spline.**

| $ratio$ | Pool | | | Factory | | |
|---|---|---|---|---|---|---|
| | PSNR↑ | SSIM↑ | LPIPS↓ | PSNR↑ | SSIM↑ | LPIPS↓ |
| 0.5 | 29.89 | 0.83 | 0.12 | 23.25 | 0.69 | 0.20 |
| 1.0 | 30.49 | 0.83 | 0.11 | 23.93 | 0.70 | 0.16 |
| 1.5 | 31.13 | 0.84 | 0.10 | 24.78 | 0.74 | 0.16 |
| 2.0 | 32.01 | 0.87 | 0.10 | 25.64 | 0.77 | 0.16 |
| 2.5 | 32.04 | 0.88 | 0.10 | 26.90 | 0.81 | 0.14 |
| 3.0 | 32.95 | 0.88 | 0.10 | 27.25 | 0.84 | 0.15 |
| 3.5 | 33.13 | 0.89 | 0.09 | 27.54 | 0.84 | 0.14 |
| 4.0 | 33.63 | 0.90 | 0.08 | 27.60 | 0.84 | 0.14 |

Table 6: **Ablation study on each module to investigate their effect on model performance.**

| Deblur | Exp. Opt. | CRF | Conti. Traj. | Factory | | | Cozyroom | | |
|---|---|---|---|---|---|---|---|---|---|
| | | | | PSNR↑ | SSIM↑ | LPIPS↓ | PSNR↑ | SSIM↑ | LPIPS↓ |
| ✗ | ✗ | ✗ | ✗ | 15.14 | 0.75 | 0.25 | 13.86 | 0.76 | 0.22 |
| ✓ | ✗ | ✗ | ✗ | 14.99 | 0.81 | 0.25 | 17.12 | 0.75 | 0.20 |
| ✗ | ✗ | ✗ | ✓ | 19.13 | 0.62 | 0.28 | 20.95 | 0.71 | 0.30 |
| ✓ | ✗ | ✗ | ✓ | 20.30 | 0.65 | 0.28 | 19.65 | 0.70 | 0.30 |
| ✗ | ✓ | ✓ | ✓ | 25.17 | 0.77 | 0.12 | 26.89 | 0.81 | 0.12 |
| ✓ | ✓ | ✓ | ✓ | 27.25 | 0.84 | 0.15 | 29.60 | 0.84 | 0.05 |

**Effect of each module.** Deblur represents the method's ability to remove blur, Exp. Opt. indicates exposure time optimization, CRF represents whether the model includes a CRF module, and Conti. Traj. refers to the use of continuous trajectories to represent camera motion. The results presented in Table 6 highlight several key findings: 1) Utilizing splines to represent the continuous camera trajectory significantly enhances model performance, achieving approximately a 24% improvement in PSNR. 2) Jointly optimizing exposure time while learning an implicit representation of the CRF substantially boosts performance, leading to a 42% increase in PSNR. This demonstrates that our method can robustly reconstruct HDR scenes in environments with varying brightness. 3) Representing motion blur as the average of a series of sharp images over the exposure time yields a 9% improvement in PSNR, showing that our approach effectively handles input images with motion blur. In summary, the proposed representation of continuous trajectories and the joint optimization of exposure time with CRF contribute significantly to the model's performance.

## 5 CONCLUSION

In this paper, we introduce a novel method **CasualHDR** for reconstructing 3D HDR scenes from videos casually captured with low-cost cameras, which often exhibit limited dynamic range and motion blur. Our method can reconstruct 3D HDR scene and generate LDR images with given specified exposures and camera poses, providing high robustness and flexibility. By leveraging the auto-exposure capabilities of modern cameras, we incorporate the high dynamic range of captured videos into a unified physical image formation model. This allows for the joint optimization with exposure time, continous-time camera trajectory, and camera response function, enabling accurate HDR scene reconstruction. Extensive experiments demonstrate that our method outperforms previous approaches in 3D HDR reconstruction.

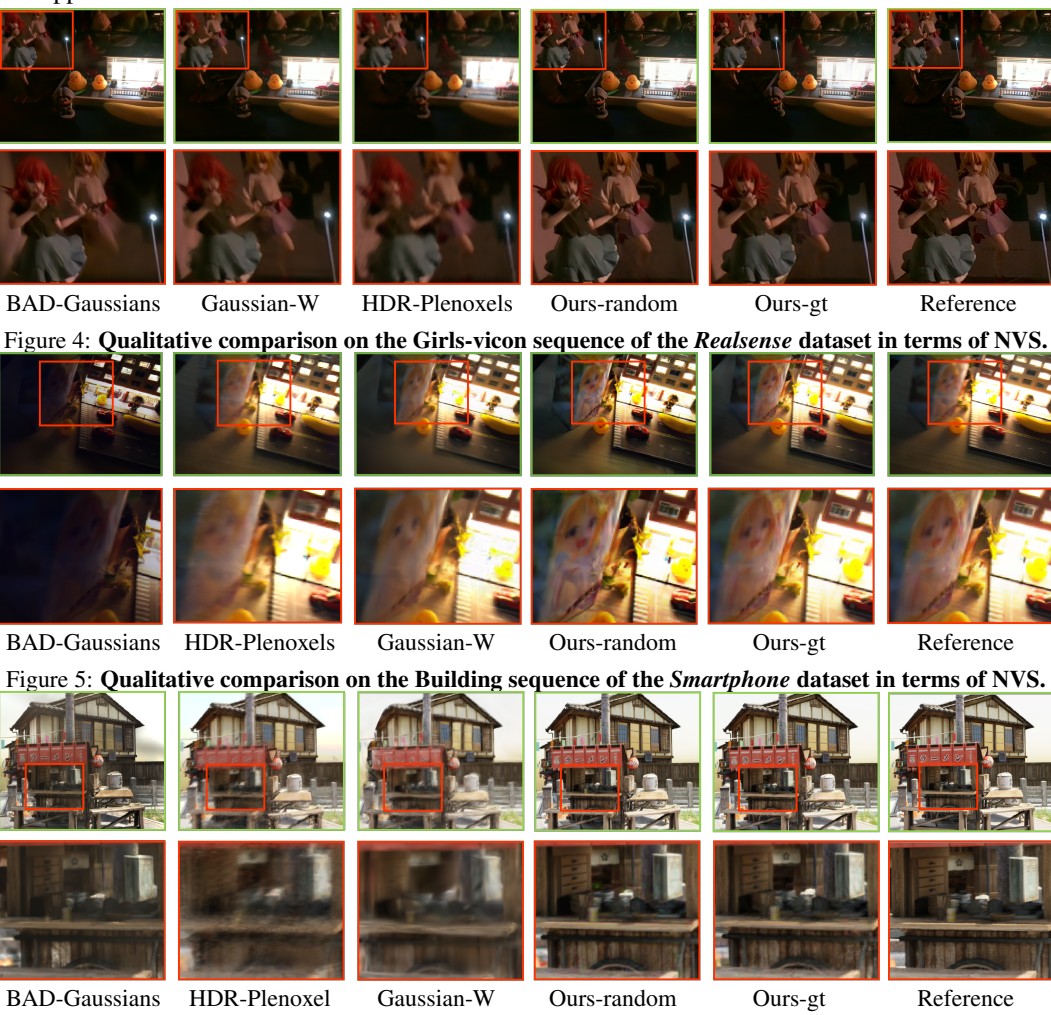

BAD-Gaussians    Gaussian-W    HDR-Plenoxels    Ours-random    Ours-gt    Reference

Figure 4: **Qualitative comparison on the Girls-vicon sequence of the *Realsense* dataset in terms of NVS.**

BAD-Gaussians    HDR-Plenoxels    Gaussian-W    Ours-random    Ours-gt    Reference

Figure 5: **Qualitative comparison on the Building sequence of the *Smartphone* dataset in terms of NVS.**

BAD-Gaussians    HDR-Plenoxel    Gaussian-W    Ours-random    Ours-gt    Reference

Figure 6: **Qualitative comparison on the Trolley sequence of the *synthetic* dataset in terms of NVS.**

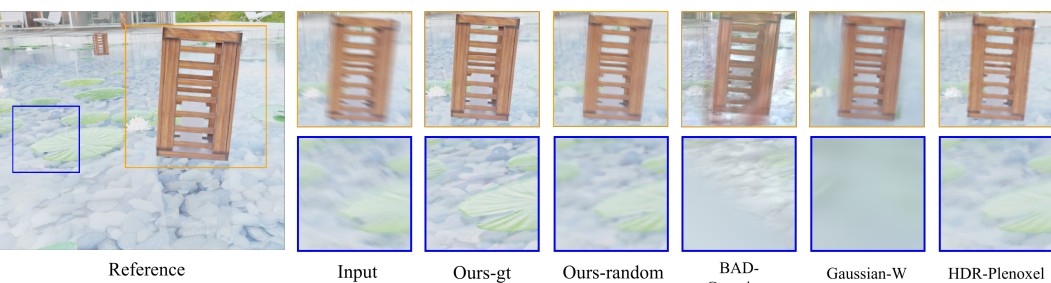

Reference    Input    Ours-gt    Ours-random    BAD-Gaussians    Gaussian-W    HDR-Plenoxel

Figure 7: **Qualitative comparison on the Pool sequence of the *synthetic* dataset under training view.BAD-Gaussians is capable to deblur the training views as ours** Due to the failure of pose optimization in the BAD-Gaussians, its image are misaligned with others.

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

## A  APPENDIX

In the appendix, we present more quantitative and qualitative experimental results for image rendering under both training and novel viewpoints. We also visualized the results of camera motion estimation and performed a qualitative comparison. The rendered novel view high frame-rate HDR video is presented in the supplementary video. We will present each part as follows.

## A.1    MORE EXPERIMENTAL RESULTS UNDER TRAINING VIEW.

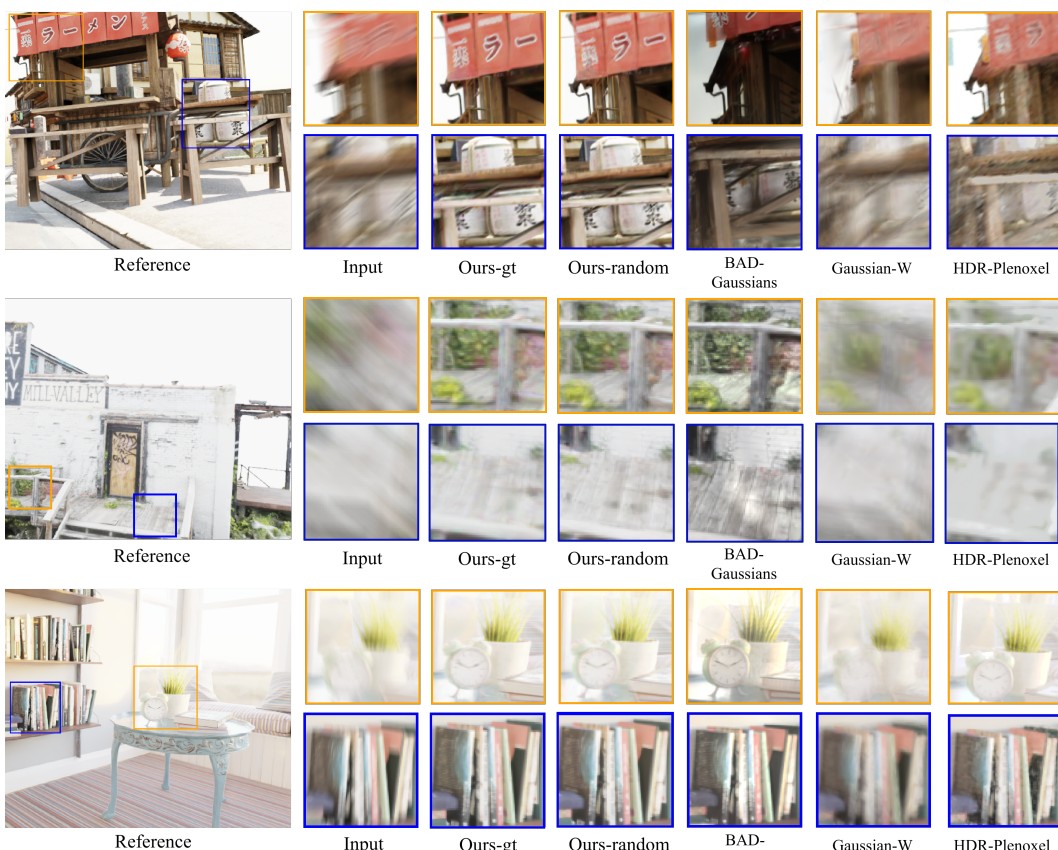

Figure 8: **Qualitative comparison on *synthetic* dataset (Trolley, Factory, Cozyroom) under training view.** BAD-Gaussians is capable to deblur the training views as ours. However, due to the failure of pose optimization in the BAD-Gaussians, its image are misaligned with others.

The results in Figure 8 demonstrate that our method effectively deblurs images under training views and achieves better image quality compared to other methods. It is worth noting that while BAD-Gaussians (Zhao et al., 2024) is also capable of deblurring images under training view, its lack of robustness to varying brightness conditions leads to pose optimization failure. As a result, The performance of deblurring is poor, even causing misalignment in the images under the training view.

Table 7: **Quantitative comparisons on the synthetic datasets in term of deblur.**

|  | Factory | | | Pool | | | Trolley | | | Cozyroom | | |
| --- | --- | --- | --- | --- | --- | --- | --- | --- | --- | --- | --- | --- |
|  | PSNR↑ | SSIM↑ | LPIPS↓ | PSNR↑ | SSIM↑ | LPIPS↓ | PSNR↑ | SSIM↑ | LPIPS↓ | PSNR↑ | SSIM↑ | LPIPS↓ |
| BAD-GS (Zhao et al., 2024) | 24.32 | 0.73 | 0.12 | 25.87 | 0.79 | 0.23 | 19.06 | 0.62 | 0.19 | 23.37 | 0.79 | 0.11 |
| BAD-GS+bilagrid (Wang et al., 2024c) | 28.25 | 0.79 | 0.08 | 31.99 | 0.86 | 0.06 | 22.16 | 0.65 | 0.15 | 26.48 | 0.81 | 0.09 |
| CasualHDR-random (ours) | 31.20 | 0.88 | 0.05 | 32.95 | 0.87 | 0.10 | 23.65 | 0.69 | 0.12 | 29.60 | 0.84 | 0.05 |
| CasualHDR-gt (ours) | 32.00 | 0.91 | 0.07 | 34.53 | 0.96 | 0.05 | 29.35 | 0.87 | 0.08 | 33.01 | 0.93 | 0.04 |

Table 8: **Quantitative comparisons on the ScanNet dataset in term of deblur (using BRISQUE metric).**

|  | scene0024_01 | scene0031_00 | scene0036_00 | scene0072_01 | scene0077_00 | scene0489_02 | Average |
| --- | --- | --- | --- | --- | --- | --- | --- |
| BAD-GS (Zhao et al., 2024) | 42.78 | 60.48 | 57.37 | 42.72 | 67.30 | 65.01 | 55.94 |
| CasualHDR-random (ours) | 38.08 | 47.65 | 53.37 | 37.55 | 62.35 | 58.23 | 49.53 |

We added comparison against the bilateral grid method (Wang et al., 2024c) applied to `gsplat` (Ye et al., 2024) and BAD-Gaussians (Zhao et al., 2024) in Table 7 and Table 10. The bilateral grids applied to NeRFs and 3DGS gives robustness to large appearance changes, enabling high quality 3D LDR reconstruction and mid-tone rendering quality. However, bilateral grids are not compatible with representing a 3D HDR scene, thus gives degraded renderings on high-contrast, over-exposed and under-exposed views.

In addition, we also evaluated the quantitative metrics for deblurring on public real-world datasets, e.g. ScanNet datastes. Due to most images of ScanNet dataset are motion-blurred, we can not find sharp reference images for evaluating, thus we utilize the no-reference image quality metric BRISQUE (Mittal et al., 2012) to quantitatively compare the deblurring performance between our method and BAD-Gaussians (Zhao et al., 2024), as shown in Table 8.

## A.2 MORE EXPERIMENTAL RESULTS UNDER NOVEL VIEW.

The quantitative experimental results in Table 9 indicate that our method significantly outperforms previous approaches in novel view synthesis on two real-world datasets. Further, the qualitative experimental results in Figure 10 and Figure 11 demonstrate that our method produces higher-quality rendered images under novel viewpoints compared to other approaches. These results indicate that our method is capable of learning accurate HDR scene representations and implicit CRF representations.

Table 9: **Quantitative comparisons on *Realsense* and *SmartPhone* dataset under novel view.**

| Method | Yakitori | | | Toufu | | |
|---|---|---|---|---|---|---|
| | PSNR↑ | SSIM↑ | LPIPS↓ | PSNR↑ | SSIM↑ | LPIPS↓ |
| gsplat (Ye et al., 2024) | 25.04 | 0.83 | 0.27 | 29.88 | 0.81 | 0.24 |
| BAD-Gaussians (Zhao et al., 2024) | 23.31 | 0.78 | 0.28 | 30.05 | 0.82 | 0.24 |
| HDR-Plenoxels (Jun-Seong et al., 2022) | 27.13 | 0.81 | 0.33 | 30.91 | 0.82 | 0.29 |
| Gaussian-W (Zhang et al., 2024) | 27.57 | 0.84 | 0.28 | 30.89 | 0.83 | 0.26 |
| CasualHDR-random (ours) | 28.56 | 0.84 | 0.22 | 32.75 | 0.87 | 0.17 |
| CasualHDR-gt (ours) | 29.19 | 0.87 | 0.16 | 32.84 | 0.91 | 0.18 |

| | Fish-pixel8pro | | | Building-pixel8pro | | | Toufu-vicon | | | Girls-vicon | | |
|---|---|---|---|---|---|---|---|---|---|---|---|---|
| | PSNR↑ | SSIM↑ | LPIPS↓ | PSNR↑ | SSIM↑ | LPIPS↓ | PSNR↑ | SSIM↑ | LPIPS↓ | PSNR↑ | SSIM↑ | LPIPS↓ |
| gsplat (Ye et al., 2024) | 23.20 | 0.82 | 0.16 | 25.99 | 0.81 | 0.11 | 24.34 | 0.81 | 0.28 | 23.81 | 0.77 | 0.28 |
| gsplat+bilagrid (Wang et al., 2024c) | 25.26 | 0.78 | 0.14 | 25.47 | 0.77 | 0.16 | 30.48 | 0.82 | 0.17 | 26.76 | 0.69 | 0.25 |
| BAD-GS (Zhao et al., 2024) | 24.28 | 0.78 | 0.14 | 26.93 | 0.82 | 0.11 | 24.22 | 0.82 | 0.24 | 23.95 | 0.77 | 0.28 |
| BAD-GS+bilagrid (Wang et al., 2024c) | 25.12 | 0.77 | 0.17 | 25.63 | 0.77 | 0.15 | 30.52 | 0.83 | 0.17 | 26.18 | 0.71 | 0.23 |
| HDR-Plenoxels (Jun-Seong et al., 2022) | 19.39 | 0.53 | 0.65 | 26.87 | 0.81 | 0.15 | 17.90 | 0.51 | 0.69 | 26.73 | 0.84 | 0.30 |
| Gaussian-W (Zhang et al., 2024) | 26.13 | 0.83 | 0.15 | 27.99 | 0.82 | 0.11 | 26.38 | 0.83 | 0.29 | 26.88 | 0.86 | 0.25 |
| CasualHDR-random (ours) | 28.30 | 0.83 | 0.13 | 28.79 | 0.83 | 0.09 | 30.87 | 0.90 | 0.15 | 32.00 | 0.90 | 0.19 |
| CasualHDR-gt (ours) | 30.81 | 0.87 | 0.12 | 29.71 | 0.85 | 0.08 | 31.34 | 0.92 | 0.12 | 32.39 | 0.91 | 0.17 |

In addition, we compare against the bilateral grid method (Wang et al., 2024c) applied to gsplat (Ye et al., 2024) and BAD-Gaussians (Zhao et al., 2024), as shown in Table 9 and Figure 9. As aforementioned, with bilateral grid (Wang et al., 2024c), BAD-Gaussians (Zhao et al., 2024) cannot represent the HDR details of the 3D scenes, thus yields degraded renderings in the high-contrast areas. As it is showed in the Figure 9, the girls in the Fish sequence of the *Smartphone* dataset are over-exposed in some views, thus exhibits under-saturation; Meanwhile, the duck in the Building sequence of the *Smartphone* dataset has lost its details and exhibits artifacts on its edge.

## A.3 MORE EXPERIMENTAL RESULTS ABOUT POSE ESTIMATION.

To demonstrate that our method can accurately recover the continuous camera motion trajectory, we visualized and compared the trajectory optimized by our method with the ground truth trajectory, as well as with other baselines. The qualitative results in Figure 12 and Figure 13 indicate that our method achieves higher pose estimation accuracy compared to previous methods.

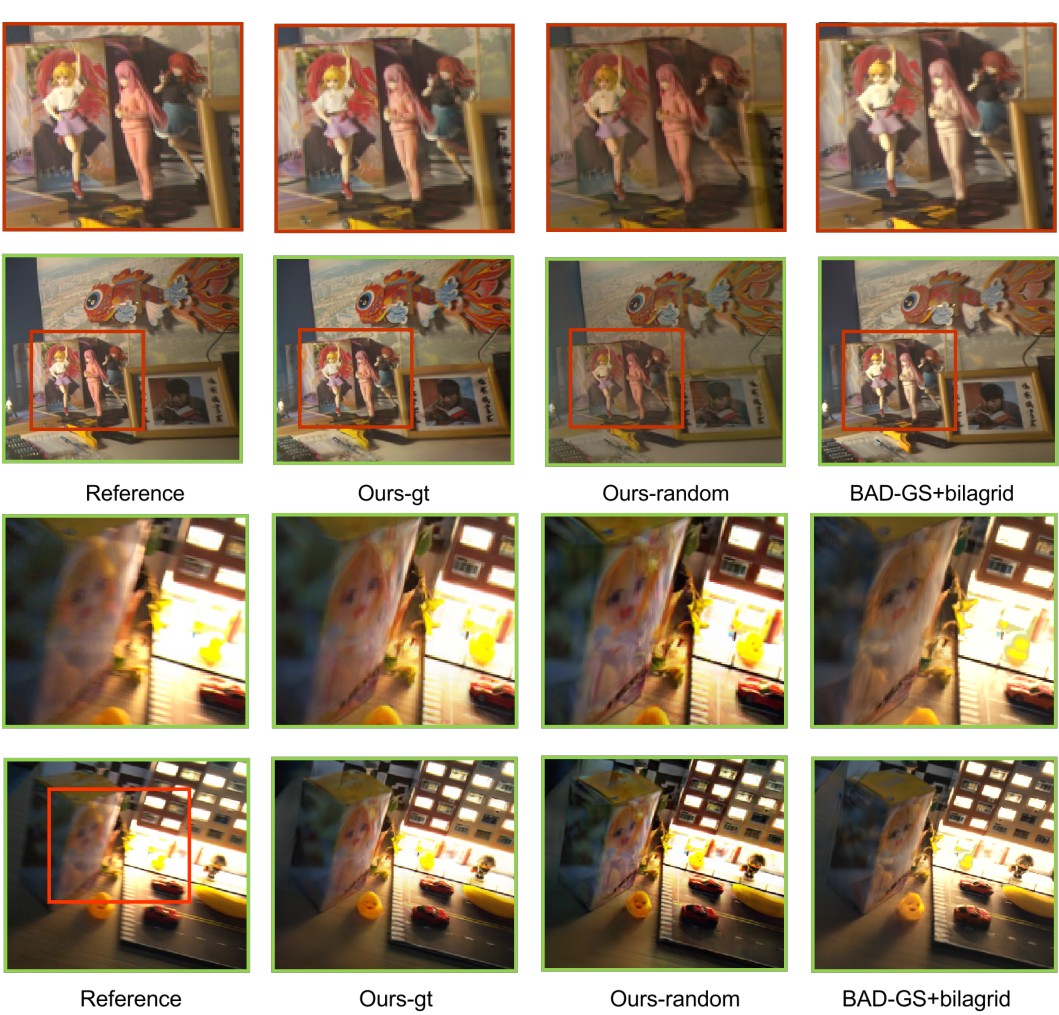

Figure 9: **Qualitative comparison with bilateral method on *Smartphone* dataset under novel view.** It is better to view the results on a monitor with high resolution and a gamut coverage close or better than sRGB.

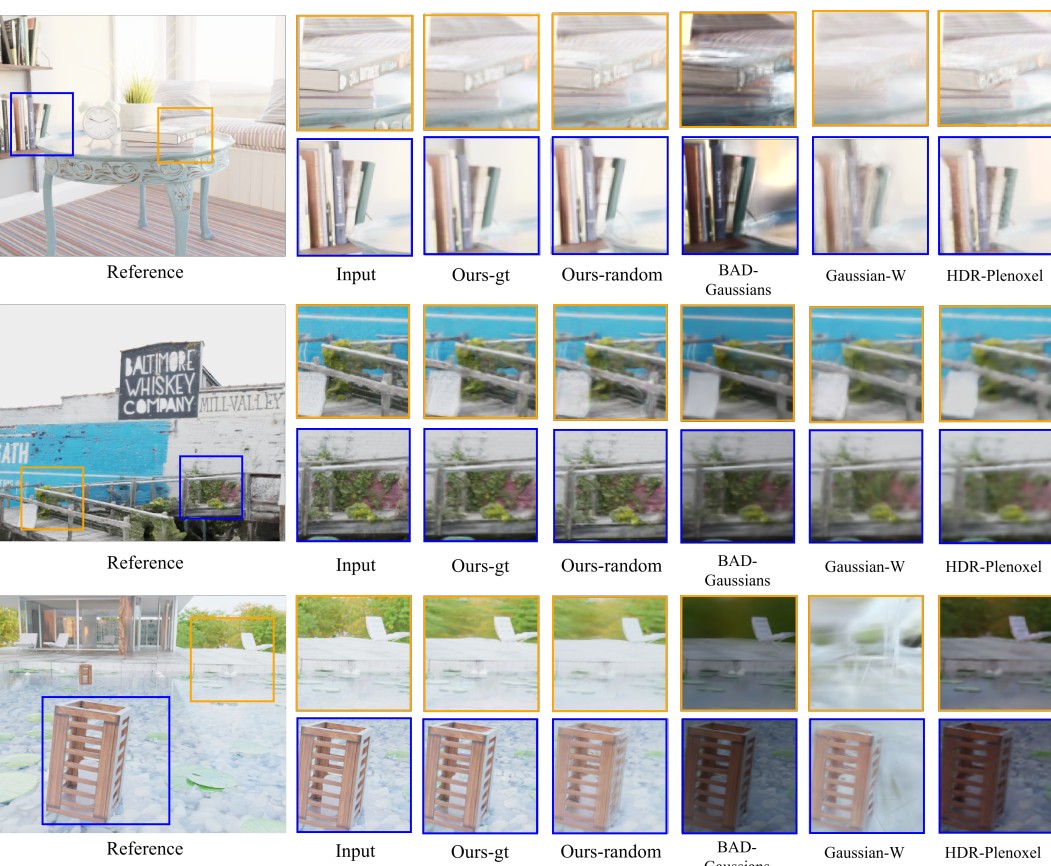

Figure 10: **Qualitative comparison on** *synthetic* **dataset(Cozyroom, Factory, Outdoorpool) under novel view.**

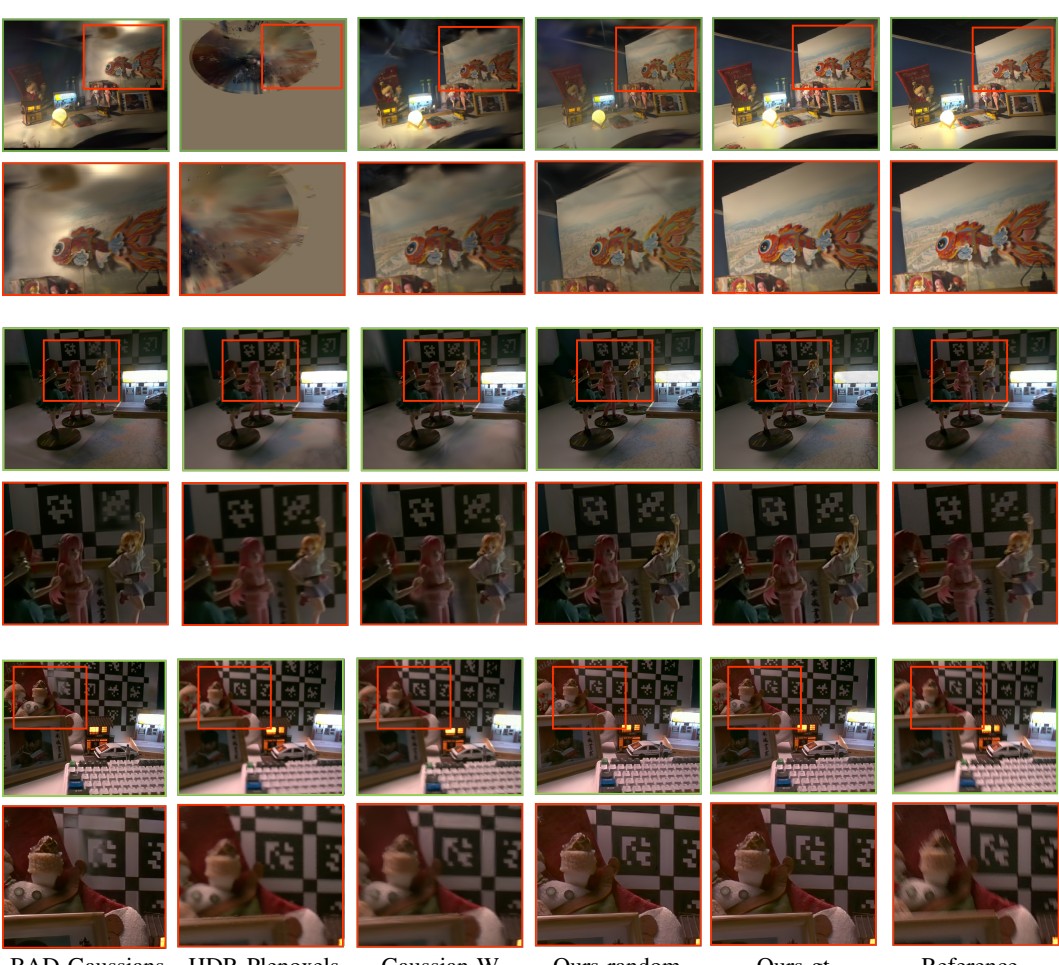

| BAD-Gaussians | HDR-Plenoxels | Gaussian-W | Ours-random | Ours-gt | Reference |

Figure 11: **Qualitative comparison on *Smartphone* dataset under novel view.**

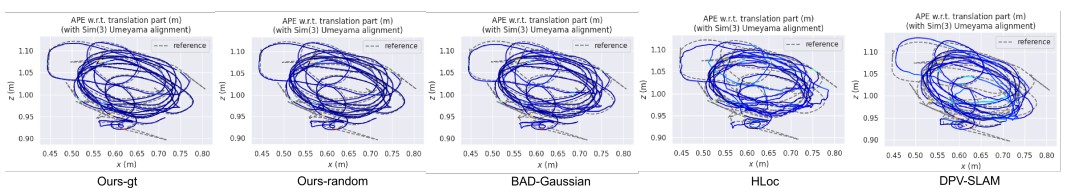

Figure 12: **Qualitative comparison for pose estimation on the Girls-vicon sequence of the *Realsense* dataset.**

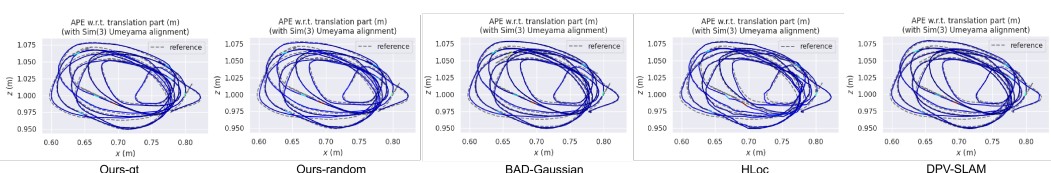

Figure 13: **Qualitative comparison for pose estimation on the Toufu-vicon sequence of the *Realsense* dataset.**

