# OpenReview forum: "CasualHDR: Robust High Dynamic Range 3D Gaussian Splatting from Casually Captured Videos"
_ICLR.cc/2025/Conference — ICLR 2025 Conference Withdrawn Submission_

### Official Review · Reviewer_QQtH · 2024-10-30

**Soundness:** 2
**Presentation:** 1
**Contribution:** 2
**Rating:** 5
**Confidence:** 5

**Summary:**

This work proposes a method, dubbed CasualHDR, for 3D HDR scene reconstruction from casual videos with auto-exposure (AE) enabled, even in the presence of severe motion blur and varying exposure time. The camera's continuous motion trajectory is introduced to initialize the 3DGS point clouds. The physical camera formation imaging model is used to simulate the motion blur and exposure change. The CasualHDR framework can jointly optimize continuous-time camera trajectory, CRF, exposure time, and 3DGS-based HDR representation.

A dataset including synthetic and real scenes is collected.

**Strengths:**

(1) The idea of jointly optimizing the camera trajectory, CRF, exposure times, and 3DGS is interesting and fancy. Because the error of the camera poses sometimes causes pixel misalignment on 2D renderings, which is usually ignore by previous methods. Besides, using the physical image formation model to jointly fit the motion blur of camera movement and exposure change is also interesting. Although using cumulative $\mathbb{SE}$(3) B-spline is a widely used continuous-time trajectory representation in robotics, it has been less studied in 3D reconstruction, especially in HDR imaging. The authors explore this part.


(2) The layout of this paper is good and clear. The teaser figure and pipeline figure are attractive. I like this style.

(3) The ablation study results in Table 6 are sufficient to validate the effectiveness of the proposed techniques. The visual comparison is also sufficient to demonstrate the advantage of the proposed method.

**Weaknesses:**

(1) The writing should be improved. Some sentences are redundant. For example, in Line 56 - 58, the authors use multiple present continuous tenses, which makes the sentence hard to read. There are also many writing typoes, such as "impoving" in Line 97 should be rectified to "improving". "$\hat{C}k(x)$" should be modified to "$\hat{C}_k(x)$". Most importantly, the method part is a mess. The first three sections seem like three independent techniques. The authors should write the relationship between them. How do they correlate and cooperate with each other? What is the pipeline? But unfortunately, I did not find this, which makes the pipeline hard to follow and incomprehensible.

(2) Some statements in this paper are confusing and even non-convincing. For instance, in Line 75 - 76, the authors claimed, " accurate exposure time is usually expensive due to the use of professional equipment". However, most of common digital cameras can easily read or just directly set the exposure time of the photos. Meanwhile, the authors themselves use the exposure time in their proposed CasualHDR. In Eq.(1), what does $\mathbf{R}_c$ represent? In Section 3.2, what does the $\mathbb{SE}(3)$ mean? It is very confusing to directly use it without any introductions.

(3) In the loss function of Eq.(8), I did not see any HDR supervision. Previous works usually use HDR supervision to constrain the rendered HDR images like HDR-NeRF. But why this work does not use HDR constraints? I check all the visual comparisons in the paper and find that the so-called HDR images have very limited luminance intensities. For example, the results in Figures 3, 4, and 5 still suffer from under- and over-exposure problems. Especially in figure 4. The details in the over-exposure area cannot be seen.  In general, the render images in this work are low-resolution and low-quality.

(4) The novelty is poor. The core technique - 3D Gaussian Splatting for HDR imaging has been explored by HDR-GS [1], which has been published by NeurIPS 2024. The techniques are similar in using tone-mapping, SfM for initialization, exposure time adapting, 3DGS rasterization, etc. However, this paper does not discuss and compare HDR-GS, which makes the contributions weaker. The motion blur estimation in the physical imaging model has been proposed and studied in Deblur-NeRF [2] and Deblur-GS [3]. However, the authors did not mention this and also did not compare with these two methods. This is not good.

[1] HDR-GS: Efficient High Dynamic Range Novel View Synthesis at 1000x Speed via Gaussian Splatting. In NeurIPS 2024.

[2] Deblur-nerf: Neural radiance fields from blurry images. In CVPR 2022.

[3] Deblur-GS: 3D Gaussian Splatting from Camera Motion Blurred Images. In ACM I3D 2024.

(5) By the way, the results of HDR editing seem like LDR results. It would be better if the main table could report the HDR and LDR results under different exposures, respectively, just like HDR-NeRF. The running time comparison is also very important.

(6) The source code and pre-trained models are not submitted. The reproducibility cannot be checked.

**Questions:**

I am curious about the results on the datasets collected by HDR-NeRF, especially the comparison (PSNR, SSIM, LPIPS, running time) with HDR-GS.

Why did you collect another dataset? Is this because of the requirement of continuous-camera trajectory?

---

> ### Author Response · Authors · 2024-11-13
> **Replies to Weakness**
>
> Thank you for your thoughtful review and valuable suggestions. We have carefully considered each point and provide our responses below.
>
> W1:We appreciate your feedback on the writing quality and clarity. We will address the identified typos and improve readability in the revised version. The core of our method is based on an innovative approach that models the camera imaging process, incorporating both the camera's motion during exposure time and the photon collection process. This significantly distinguishes our work from methods like HDR-NeRF and HDR-GS, which do not account for camera motion, making them more sensitive to motion blur. Additionally, unlike HDR-NeRF and HDR-GS, which require ground truth exposure time for 3D HDR scene reconstruction, our approach uses continuous trajectories to estimate the initial exposure time through bundle adjustment, which is subsequently optimized. This flexibility allows our method to reconstruct HDR scenes without ground truth exposure time, enhancing its adaptability.
>
> W2:For users relying on mobile devices rather than digital cameras, exposure data can be challenging to obtain. While we could access this information using low-level camera APIs on a Pixel Pro 8 for our dataset, only limited free software provides this feature. Furthermore, with video inputs where exposure time can vary across frames (especially under auto-exposure settings), commonly used formats (e.g., MP4) do not store per-frame exposure times. What's more, SE(3) is a foundational concept in robotics, representing the space in which poses are optimized, and it is typically not further explained in related works. However, we will ensure that it is clearly defined in the paper to avoid confusion for readers less familiar with this concept.
>
> W3:We understand there may have been a misunderstanding here. HDR-NeRF does not directly employ HDR supervision but instead uses photometric consistency with LDR images, along with unit exposure loss to constrain the radiance range. In our approach, we similarly supervise using LDR images with photometric consistency. However, due to the SH representation of 3D Gaussian Splatting (3DGS) limiting radiance to a range of 0–1, we do not require a unit exposure loss. The 3D HDR radiance field reconstructed by our method indeed has a wide dynamic range, but due to display limitations, the images in the paper are rendered as LDR images with specified exposure times. The over- and under-exposure results you mentioned demonstrate our method's capability to reconstruct the 3D HDR scene, from which images with varying artistic effects can be rendered by adjusting exposure time. Figure 3 demonstrates how we can reveal more detail in high-contrast areas by adjusting exposure.( it seems there may be some misunderstanding between LDR/HDR and under/over-exposure, The difference between LDR and HDR images lies in the range of scene radiance they capture. Overexposure and underexposure, however, are more related to human perception of the image. )

---

> > ### Author Response · Authors · 2024-11-13
> > **Replies to Weakness(W4 - )**
> >
> > W4:We appreciate your interest in our approach’s novelty. We have detailed the differences between HDR-GS and our method in both the paper’s introduction and above. While both methods aim to reconstruct 3D HDR scenes, our approach offers greater flexibility and robustness, as demonstrated in our experiments. While tone mapping and exposure adaptation are common across HDR methods,SfM for initialization,  3DGS rasterization are common across 3DGS methods,our distinct contribution lies in integrating camera motion during exposure time, enhancing flexibility, cost-effectiveness, and robustness.
> >
> > W5:It seems there may be some misunderstanding between us. Our work involves HDR scene reconstruction, but visualizing an HDR scene requires tone mapping. By setting a specific exposure time, we define a dynamic range that is then converted into an LDR image for display on the screen. This approach is consistent with HDR-NeRF and HDR-GS, as you can verify by closely checking their results.
> >
> > W6:Since HDR-GS code is unavailable, direct comparisons are challenging. Deblur-NeRF, which uses a sparse kernel for deblurring, has limited relevance to camera imaging. Deblur-GS and BAD-GS share a methodology, and our experiments demonstrate our method’s advantages over BAD-GS. As for the source code and pre-trained models, we plan to release them post-review to maintain confidentiality.
> >
> > Q:HDR-NeRF’s dataset comprises individual images captured at unique camera poses rather than video sequences, differing from our target input. However, as per your suggestion, we plan to add our results on this dataset. We would also like to emphasize that our focus is on HDR scene reconstruction from casually captured videos. Since existing datasets did not meet our requirements, we created a custom dataset. For comparison, we included results on ScanNet due to its similar capture conditions, with more details available in the supplementary materials.
> >
> > Thank you again for your constructive comments and questions. We hope these responses address your concerns and provide additional clarity regarding our contributions.

---

### Official Review · Reviewer_RqLZ · 2024-11-01

**Soundness:** 2
**Presentation:** 3
**Contribution:** 2
**Rating:** 3
**Confidence:** 4

**Summary:**

The authors of this paper propose a new problem: how to recover an HDR scene from a blurry LDR videos captured with a handheld camera under different exposures.

**Strengths:**

1. The authors propose a novel problem: how to recover an HDR scene from casual LDR videos that exhibit ghosting blur.
2. To validate their method, the authors created their own dataset and conducted detailed experiments on it.

**Weaknesses:**

1. The authors do not propose a new method; they simply combine existing approaches. While this does present certain challenges, it is not novel.
2. The authors claim that their method is an HDR reconstruction method; however, I did not find any HDR measurement results, such as HDR-VDP, PUPSNR.
3. Line 36, "Prior works have focused on high dynamic range (HDR) scene recovery, typically require repeatedly capturing of multiple sharp images with different exposure times at fixed camera positions, which is time-consuming and challenging in practice" This statement raises a question, as current HDR reconstruction methods do not require capturing multiple clear images at fixed camera positions with different exposure times. For example, HDRNeRF can reconstruct an HDR radiance field using just a single image from 18 different viewpoints.
4. The teaser illustration contains errors: xxx(ours).
5. Providing the videos may be more convincing.

**Questions:**

1. Why did you choose to model the CRF using an MLP instead of an explicit method? An explicit method would likely be faster.

---

> ### Author Response · Authors · 2024-11-13
>
> Thank you for your insightful review and valuable suggestions. We carefully considered each weakness and provide our responses below.
>
> **W1**. We thank the reviewer for spending your time and the thoughtful comments. We appreciate the reviewer for the acknowledgment of our discovery of the novel problem: how to recover an HDR scene from casual LDR videos that exhibit ghosting blur. However, we kindly raise our disagreement with the reviewer's opinion that we do not propose a new method. We do draw inspiration from pioneering works like BAD-NeRF (mitigate motion blur by joint optimization of camera motion and 3D scene) and HDR-NeRF (recover 3D HDR scene from 2D LDR images) to tackle this novel problem. However, a direct combination of current methods does not yield satisfactory results, as we demonstrated in our ablation study. Through investigation into the physical imaging process of casual videos, we found that the inconsistency of brightness, as well as the motion blurring, can both be unified into a single model, that simultaneously considers the varying exposure time, and the camera’s motion during the exposure time (i.e. the photon collection process). This is a significant distinction from methods such as HDR-NeRF and HDR-GS, which do not account for camera motion and, consequently, are highly sensitive to motion blur in images. Moreover, unlike HDR-NeRF and HDR-GS, which require ground truth exposure time to reconstruct 3D HDR scenes, our approach leverages continuous trajectories to obtain an initial estimate of exposure time through bundle adjustment, which is further optimized during processing. Thus, our method allows for HDR scene reconstruction without the need for ground truth exposure time, providing greater flexibility compared to other approaches.
>
> **W2**. Thank you for your professional advice on the quantitative evaluation. In our experiments, we followed the evaluation protocols of HDR-NeRF and HDR-Plenoxels. We will add the metrics you asked to our quantitative experiments.
>
> **W3**. Thank you for your in-depth investigation. However, there seems to be a misunderstanding in our Abstract. What we are trying to present is that, current 3D HDR reconstruction methods use multiple LDR images at fixed positions, with different exposure times to reconstruct the 3D HDR scene as well as estimate the CRF function, while in our casual HDR task, the camera position is not fixed, thus will face motion blur during exposure time, just like we explained in our Introduction. Thus, the key challenge is reducing the cost of data acquisition, enabling high-quality 3D HDR scene reconstruction with casual capturing with consumer-grade devices. We will improve our presentation in the abstract.
>
> **W4**. Thank you for your carefulness. We will correct the error in our teaser.
>
> **W5**. Thank you for your valuable suggestions. We will provide a video to better demonstrate the effectiveness of our method.

---

> > ### Comment · Reviewer_RqLZ · 2024-11-14
> >
> > Thank you for the author’s response. Although I acknowledge the substantial contributions made, I find the innovation somewhat lacking. It seems the author is addressing an engineering problem rather than introducing new techniques. For this reason, I feel that this paper may not be a good fit for the ICLR community, which is why I am inclined to recommend rejection. Additionally, I sense that the author may have some misunderstandings about this field:
> >
> > W3: "current 3D HDR reconstruction methods use multiple LDR images at fixed positions".  In fact, methods like HDRNeRF can reconstruct HDR radiance field with only 18 viewpoints, each with just one randomly selected exposure.

---

> ### Author Response · Authors · 2024-11-14
>
> Thank you very much for your thoughtful reply and insights. We truly appreciate your feedback on our work's contributions, and we understand your perspective regarding the expectations of the ICLR community. We would, however, like to clarify our view on the novelty and technical advancements in our approach, which we believe contribute substantial improvements to the robustness and rendering quality of 3DGS.
>
> To outline our main contributions more clearly:
>
> 1, **Continuous-Time Trajectory Constraint:** We explore continuous-time trajectory constraints in the reconstruction of 3DGS, an area we have not seen addressed in the literature.
>
> 2, **Bundle Adjustment on Continuous-Time Trajectories:** We developed an approach for conducting bundle adjustment on continuous trajectories, enabling effective image deblurring and exposure time optimization—techniques, to our knowledge, not previously implemented in this context.
>
> 3, **Unified Neural Imaging Model:** Through these explorations above, we introduce a unified differentiable imaging model that can jointly estimate exposure time and image blur, greatly enhancing both the robustness and flexibility of 3DGS.
>
> Regarding 'W3,' we acknowledge and agree with your point that methods like HDRNeRF can indeed reconstruct HDR radiance fields with as few as 18 viewpoints, each using one randomly selected exposure. In our previous response, we intended 'fixed' to refer to a stationary camera pose during the exposure time, while 'multiple'(multi-view) highlights the need for more than one image to reconstruct the entire scene (as you mentioned 18 viewpoints). Our aim was to convey that capturing numerous photos with the camera stationary (typically requiring a tripod) during exposure time can be challenging and time-consuming, especially compared to the convenience of a casually captured video sequence. For a more detailed explanation, please refer to lines 74–80 in the introduction of our paper, our experiments also follow this setting.
>
> Thank you once again for the opportunity to address these points, and we sincerely appreciate your constructive feedback.

---

### Official Review · Reviewer_qhC6 · 2024-11-03

**Soundness:** 3
**Presentation:** 3
**Contribution:** 3
**Rating:** 6
**Confidence:** 4

**Summary:**

The paper introduces CasualHDR, a high dynamic range (HDR) scene reconstruction method based on 3D Gaussian Splatting (3DGS). This approach can reconstruct 3D HDR scenes from casually captured videos, even if they include automatic exposure adjustments and motion blur. CasualHDR relies on a physical image formation model to jointly optimize exposure time, the camera response function (CRF), continuous camera motion trajectory, and HDR scene representation. The authors conducted extensive experiments, showing advantages in image quality and robustness, and demonstrated the potential of this method for applications like novel view synthesis, deblurring, and HDR editing.

**Strengths:**

1. CasualHDR does not require precise exposure times as input, which significantly reduces data collection costs and improves adaptability to various exposure conditions and blurry situations.
2. This method shows good image quality and can produce realistic HDR reconstructions even in scenes with uneven exposure and extreme lighting changes.
3. The method demonstrates versatility, including applications in novel view synthesis, deblurring, and exposure adjustment, showing potential value in real-world scenarios.
4. Extensive experiments on multiple synthetic and real datasets show its performance in image quality and positional accuracy, proving the method's practicality and robustness.

**Weaknesses:**

1. ***Limited novelty***: The method combines 3D Gaussian Splatting (3DGS) with a physical image formation model, mainly adapting HDR-NeRF’s MLP tone mapping for CRF handling and BAD-Gaussians' camera motion modeling. Although this integration is effective, it largely builds on existing techniques with limited original innovation. While CasualHDR’s addition of exposure time optimization could enhance flexibility in varying lighting, the lack of error analysis for these estimates raises concerns about the accuracy and robustness of this component.

2. ***High computational cost***: The method processes CRF on a per-pixel basis, which significantly increases computation, especially with high-resolution images or large datasets, potentially limiting its applicability in real-time scenarios.

3. ***Accuracy of exposure estimation***: Although the authors claim the optimization effectively estimates exposure time, they do not provide detailed error analysis or comparisons with ground truth exposure times, which raises questions about its robustness.

**Questions:**

1. Given that CRF is applied on a per-pixel basis, which increases computation time significantly for high-resolution or large-scale images, could the authors provide details on the training time and rendering speed for exposure editing? Additionally, are there optimization strategies to address this?

2. Could the authors include an error analysis comparing the optimized exposure times with the ground truth to demonstrate the effectiveness of the exposure time optimization?

3. Could the authors provide comparisons of the learned CRF curves with ground truth or typical CRF shapes from the literature to illustrate the effectiveness of the CRF modeling?

---

> ### Author Response · Authors · 2024-11-15
>
> Thank you for your appreciation!

---

### Note · Authors · 2024-11-15

I have read and agree with the venue's withdrawal policy on behalf of myself and my co-authors.